# COVID-19 Vaccine and Death: Causality Algorithm According to the WHO Eligibility Diagnosis

**DOI:** 10.3390/diagnostics11060955

**Published:** 2021-05-26

**Authors:** Cristoforo Pomara, Francesco Sessa, Marcello Ciaccio, Francesco Dieli, Massimiliano Esposito, Giovanni Maurizio Giammanco, Sebastiano Fabio Garozzo, Antonino Giarratano, Daniele Prati, Francesca Rappa, Monica Salerno, Claudio Tripodo, Pier Mannuccio Mannucci, Paolo Zamboni

**Affiliations:** 1Department of Medical, Surgical and Advanced Technologies “G.F. Ingrassia”, University of Catania, 95121 Catania, Italy; massimiliano.esposito91@gmail.com (M.E.); monica.salerno@unict.it (M.S.); 2Department of Clinical and Experimental Medicine, University of Foggia, 71122 Foggia, Italy; francesco.sessa@unifg.it; 3Clinical Molecular Medicine and Laboratory Medicine, Institute of Clinical Biochemistry, Department of Biomedicine, Neurosciences and Advanced Diagnostics, University of Palermo, 90127 Palermo, Italy; marcello.ciaccio@unipa.it; 4Department of Laboratory Medicine, AOUP “P. Giaccone”, 90127 Palermo, Italy; 5Central Laboratory of Advanced Diagnosis and Biomedical Research (CLADIBIOR), University of Palermo, 90128 Palermo, Italy; francesco.dieli@unipa.it; 6Department of Health Promotion, Mother and Child Care, Internal Medicine and Medical Specialties (Pro.M.I.S.E.), University of Palermo, 90128 Palermo, Italy; giovanni.giammanco@unipa.it; 7Clinical Pathology Unit, Garibaldi Centro Hospital, ARNAS Garibaldi, 95121 Catania, Italy; fgarozzo41@gmail.com; 8Department of Surgical, Oncological and Oral Science (Di.Chir.On.S.), University of Palermo, 90128 Palermo, Italy; antonino.giarratano@unipa.it; 9Department of Anesthesia, Intensive Care and Emergency, Policlinico Paolo Giaccone, 90128 Palermo, Italy; 10Fondazione IRCCS Ca’ Granda Ospedale Maggiore Policlinico, Department of Transfusion Medicine and Hematology, 20162 Milan, Italy; daniele.prati@policlinico.mi.it; 11Department of Biomedicine and Neurosciences and Advanced Diagnostics, University of Palermo, 90127 Palermo, Italy; francesca.rappa@unipa.it; 12Tumor Immunology Unit, Department of Health Sciences, Human Pathology Section, University of Palermo School of Medicine Palermo, 90128 Palermo, Italy; claudio.tripodo@unipa.it; 13Fondazione IRCCS Ca’ Granda Ospedale Maggiore Policlinico, Angelo Bianchi Bonomi Hemophilia and Thrombosis Center, 20162 Milan, Italy; piermannuccio.mannucci@policlinico.mi.it; 14Hub Center for Venous and Lymphatic Diseases Regione Emilia-Romagna, Sant’Anna University Hospital of Ferrara, 44124 Ferrara, Italy; zambo@unife.it

**Keywords:** SARS-CoV-2, COVID-19, vaccine, immune thrombocytopenia, disseminated intravascular coagulation, deep vein thrombosis, vaccination campaign, standard protocol, autopsy, post-mortem investigation

## Abstract

The current challenge worldwide is the administration of anti-severe acute respiratory syndrome coronavirus 2 (SARS-CoV-2) vaccines. Even if rarely, severe vascular adverse reactions temporally related to vaccine administration have induced diffidence in the population at large. In particular, researchers worldwide are focusing on the so-called “thrombosis and thrombocytopenia after COVID-19 vaccination”. This study aims to establish a practical workflow to define the relationship between adverse events following immunization (AEFI) and COVID-19 vaccination, following the basic framework of the World Health Organization (WHO). Post-mortem investigation plays a pivotal role to support this causality relationship when death occurs. To demonstrate the usefulness and feasibility of the proposed workflow, we applied it to two exemplificative cases of suspected AEFI following COVID-19 vaccination. Based on the proposed model, we took into consideration any possible causality relationship between COVID-19 vaccine administration and AEFI. This led us to conclude that vaccination with ChAdOx1 nCov-19 may cause the rare development of immune thrombocytopenia mediated by platelet-activating antibodies against platelet factor 4 (PF4), which clinically mimics heparin-induced autoimmune thrombocytopenia. We suggest the adoption of the proposed methodology in order to confirm or rule out a causal relationship between vaccination and the occurrence of AEFI.

## 1. Introduction

The current challenge worldwide is the administration of anti-severe acute respiratory syndrome coronavirus 2 (SARS-CoV-2) vaccine. To date, the European Community, following the recommendation by the European Medicines Agency (EMA), has authorized the use of four vaccines. The vaccine of the company BNT162b2 (Pfizer–BioNTech) was authorized on December 21, 2020 [1], the second mRNA-1273 (Moderna) was approved on January 6, 2021 [2]; the third vaccine, ChAdOx1 nCov-19 (AstraZeneca), was approved on January 29, 2021 [3]; the last vaccine is COVID-19 Vaccine Janssen (Johnson & Johnson), authorized on March 11, 2021 [4]. Many other vaccines are currently ongoing clinical trials [5].

Nevertheless, severe adverse reactions temporally related to vaccine administration have generated diffidence in the population, slowing the European vaccine plan. On April 7, 2021, the EMA admitted a possible link with very rare cases of unusual blood clots with low platelets after the administration of ChAdOx1 nCov-19 (formerly COVID-19 Vaccine AstraZeneca) [6]. For this reason, the scientific community is working to support the concept of safer COVID-19 vaccinations. Moreover, researchers worldwide are focusing their attention on the so-called “thrombosis and thrombocytopenia after ChAdOx1 nCoV-19 vaccination”, producing several studies and publications [7,8,9]. Furthermore, after a precautionary stop, on April 20, 2021, the COVID-19 vaccine Janssen received the authorization for its use by EMA that provided updated guidance, confirming that the overall benefit–risk profile remains positive. However, it has been confirmed that a small number of vascular adverse events involving blood clots in combination with low platelet counts may occur within approximately one to three weeks following vaccination [10].

This study aims to establish a practical workflow to define the causal relationship between adverse events following immunization (AEFI) and COVID-19 vaccination, following the basic framework of the World Health Organization (WHO) [11]. Post-mortem investigation plays a pivotal role to support this causality relationship when the death occurs. To demonstrate the usefulness and feasibility of the proposed workflow, we applied it to two exemplificative cases of suspected AEFI following COVID-19 vaccination.

## 2. Materials and Methods

To develop a practical workflow to define the relationship between AEFI and COVID-19 vaccination following immunization, we worked in two steps: (i) application of the global WHO guidelines on two fatal cases temporarily related to COVID-19 vaccine administration who underwent forensic autopsy; (ii) development of a practical workflow to be adopted in similar cases to collect important information on adverse events following vaccination.

### 2.1. Structure of the WHO AEFI Guidelines

First of all, it is important to apply the mandatory four steps for causality assessment for each AEFI case: eligibility, checklist, algorithm, and classification.

In the first step, “eligibility”, it is important to ascertain that the vaccine is administered before the event. This can be established by obtaining a very detailed and careful clinical history and physical findings plus collecting information from relevant informer. It is also crucial to have a valid diagnosis for the referred AEFI, which could be an adverse or unwanted sign, an outlier laboratory result, a symptom or disease. Thus autopsy should be considered mandatory as the gold standard method in medicine to identify the exact cause of death. At the end of this step, the causality question is usually formulated. For example, evaluating the AEFI after COVID-19 vaccination, the question is if the COVID-19 vaccine caused thrombocytopenia.

After these steps, it is important to proceed with the checklist, answering the following questions:-Is there strong evidence for other causes? To answer this question, it is fundamental to analyze the medical history of the patient, focusing on the clinical examination to confirm the relationship.-Is there a known causal association with the vaccine or vaccination? For this question, is important to analyze the vaccine product and vaccine quality. Moreover, it is important to exclude immunization errors and immunization stress-related responses.-Is there strong evidence against a causal association? At this point, a literature review should be performed in order to exclude the presence of published evidence against a causal association between vaccine administration and the event.-Finally, it is important to analyze other qualifying factors for classification (for example, pre-existing conditions, event-related to previous vaccinations, etc.).

If the relationship between the two events persists, the AEFI could undergo the algorithm reported in the WHO document. Following this algorithm, it will be possible to classify the AEFI. In particular, four categories are identified in the section classification:A.Consistent with causal association to immunization;B.Indeterminate;C.Inconsistent with causal association to immunization;D.Unclassifiable.

### 2.2. Adaptation of the Global WHO Guidelines to AEFI Post COVID-19 Vaccine

Starting from the WHO version of the general guidelines summarized in Section 2.1, we developed a set of items tailored to AEFI that occurred post-COVID-19 vaccine.

### 2.3. Vaccine Characteristics and Exhaustive Review of the Literature

First of all, we analyzed the characteristics of each authorized vaccine, focusing on its adverse effects (Appendix A). We have particularly analyzed the interim recommendations of WHO for the use of the Pfizer–BioNTech COVID-19 vaccine [12], Moderna mRNA-1273 vaccine [13], the ChAdOx1 nCoV-19 (ChAdOx1-S (recombinant)) vaccine developed by Oxford University and AstraZeneca [14], and Janssen Ad26.COV2.S (J&J) [15].

Moreover, in order to develop specific evidence-based items, an exhaustive review of the pertinent literature was performed. To date, several reports have been published reporting severe adverse effects (thrombosis and thrombocytopenia) after COVID-19 vaccination [7,8,9,16,17,18,19]. Each report was analyzed independently by each author of this article in order to identify specific items.

We then qualitatively synthesized evidence and references in the items in the checklist. Moreover, we reviewed the first draft with the help of two clinicians and modified it based on their feedback. The second draft was analyzed by two epidemiologists and following their independent reports, we adapted it. Finally, the third version was used in a pilot case, analyzing all the feedback to complete its final form.

### 2.4. Exemplificative Cases Needing AEFI Diagnosis after ChAdOx1 nCoV-19 Vaccine Administration

Two cases (case 1, 1 male, 50 y.o.; case 2, 1 female, 37 y.o.) of suspected AEFI temporarily related to COVID-19 vaccine administration are presented in this report to explain the causal assessment. Both subjects died after COVID-19 vaccine administration, consisting of a genetically modified adenoviral vectors (ChAdOx1 nCoV-19). Before the autopsy, both medical records were carefully consulted. Moreover, family medical histories were obtained from the respective general practitioners. Furthermore, before each autopsy, a total body CT scan was performed to confirm the absence of tumors. All scans were performed with the body in a supine position using a helical 16-slice CT scanner (Philips Ct Brilliance 16).

This finding was confirmed by the detailed post-mortem examination of all organs. In case 1, the patient suffered only from obstructive sleep apnea. The autopsy was performed 24 h after death. In case 2, personal medical history was nil. Autopsy was performed 3 days after death; the corpse was kept at −4 °C to avoid post-mortem modifications. All biological fluids collected during hospitalization were properly stored and used to perform all tests. All procedures performed in the study were approved by the scientific committee of the Department of Medical and Surgical Sciences and Advanced Technologies “G.F. Ingrassia”, University of Catania, (record n. 21/2020) and were performed in accordance with the 1964 Helsinki Declaration and its later amendments or comparable ethical standards. The prosecutor authorized the use of anonymous data according to Italian law. No informed consent is required to use information from deceased persons when such information is indispensable and relevant for scientific and research purposes.

In case 1, a man who suffered from abdominal pain 10 days after ChAdOx1 nCoV-19 vaccination was admitted to the emergency department with severe thrombocytopenia, low plasma fibrinogen, and very high levels of D-dimer. CT showed occlusive portal vein thrombosis with smaller thrombi in the splenic and upper mesenteric veins. Clinical conditions deteriorated quickly: a new CT scan showed a massive intracerebral hemorrhage. After various therapies, the patient died 4 days after the onset of symptoms and 16 days after vaccination.

In case 2, a woman suffered from strong low back pain and headache 10 days post-vaccination with ChAdOx1 nCoV-19. In the early morning of day 11, she was found to be unconscious, and was transferred to the emergency department. The blood parameters were similar to those of case 1 and the CT scan showed a very large intracranial hemorrhage and the presence of an occlusive thrombus in the superior sagittal sinus. She underwent craniotomy in order to reduce intracranial hypertension and remove the frontal hemorrhage, but at the end of the intervention, she remained comatose. Her clinical conditions worsened and she died 24 days after vaccine administration.

A complete post-mortem investigation was performed in both cases, carrying out histological and immunohistochemical (IHC) investigations. IHC staining was carried out using the Novolink Polymer Detection Systems (Leica Novocastra) or Bond Polymer Refine Detection Kit (Leica Novocastra) and DAB (3,3′-Diaminobenzidine, Novocastra) as substrate chromogen. The following antibodies were used: Anti-CD163 (clone 10D6, cod. PA0090, Leica Biosystems NewCastle Ltd), Anti-CD66b (clone BY114, cod. AM325, BioGenex), Anti-C1r (cod. HPA001551, Sigma-Aldrich), Anti-C4d (cod. 404A, Cell Marque), Anti-IgM (clone 8H6, cod.PA0278, Leica Biosystems NewCastle Ltd., Newcastle upon Tyne NE12 8EW, UK), Anti-IgG (clone RWP49, cod. PA0905, Leica Biosystems NewCastle Ltd., Newcastle upon Tyne NE12 8EW, UK).

We applied the proposed flowchart to these exemplificative cases related to a single COVID-19 vaccine, even though we recommend to apply it to each severe AEFI related to COVID-19 vaccination.

### 2.5. Autopsy Methodology

Autopsy should be performed following the international guidelines [20,21] for post-mortem investigation of suspected or confirmed cases of people dead with/from COVID-19 [22,23,24]. A negative molecular swab during life does not exclude the presence of SARS-CoV-2 in the lower respiratory airways [25]. Thus, we performed a tissue evaluation of a molecular test for SARS-CoV-2. Autopsy of the two exemplificative cases were performed according to the Letulle technique recommended for clinical and forensic assessment in case of suspected death related to vaccine administration [26]. Before tissue fixation, we collected fresh tissues from all organs, inserting each sample in cassettes containing RNA-stabilizing solution, and storing them at −20 °C. For the first time, we adopted the protocol established by Sicily’s Regional taskforce for the implementation of diagnostic findings and autopsy examinations in COVID-19 positive corpses or in the event of post-vaccination deaths [27]

Biological fluids were also collected, such as blood (usually from the iliac vein or right atrium), urine, and bile. We harvested 2 samples of cadaveric blood, and 1 sample of fresh tissue (preferably spleen, to store at −20 °C) to perform the genetic tests for hereditary thrombophilia [28,29,30,31,32,33,34]). To confirm the absence of a myeloproliferative neoplasm, a bone marrow biopsy at the left iliac crest was performed before the autopsy examination. Moreover, it is likely to be excluded by the negative history thereof and normal blood counts in repeated tests routinely done before the event in both subjects.

### 2.6. Tests and Biomarkers

A complete set of histological and immunohistochemistry (IHC) investigations were also performed on the harvested tissues. The histopathology and pathological characteristics were analyzed by standard hematoxylin and eosin (H&E) staining following the standard protocol [35]. Moreover, we performed the IHC investigations in order to characterize, by means of specific biomarkers, the inflammatory cells involved (particularly CD163, and CD66b); to verify the presence of adhesion molecules (anti-VCAM1 antibody) and the activation of the complement pathway (anti-C1r antibody, anti-C4d antibody); the deposition of antibodies both of IgM and IgG classes and case-specific antibodies (platelet factor 4 PF4/heparin complex).

Furthermore, we performed the following tests when these were not performed during the hospitalization period:Thrombosis panel test (D-dimer, prothrombin time and international normalized ratio (PT/INR), partial thromboplastin time (PTT, aPTT), complete blood count (CBC), antiphospholipid antibodies, lupus anticoagulant testing, antithrombin, protein C, protein S, and ID-Heparin/PF4 antibody test).Genetic testing for thrombophilia (factor V Leiden mutation, factor II 20210 mutation).Molecular tests for detection of SARS-CoV-2 using other lung samples (i.e., bronchial wash (BW)/bronchoalveolar lavage (BAL) specimens, post-mortem lung swab).Molecular tests for detection of SARS-CoV-2 in atypical samples, using formalin-fixed paraffin-embedded (FFPE) tissue specimens.Panel for respiratory viruses (adenovirus, coronavirus (229E, HKU1, NL63, OC43), human metapneumovirus, human rhinovirus/enterovirus, influenza A, influenza A H1, influenza A H1-2009, influenza A H3, influenza B, parainfluenza 1, parainfluenza 2, parainfluenza 3, parainfluenza 4, respiratory syncytial virus A, respiratory syncytial virus B, chlamydia pneumonia, mycoplasma pneumonia).Panel for respiratory bacterial pathogens (*Bordetella pertussis, Chlamydophyla pneumoniae, Mycoplasma pneumoniae, Legionella pneumophila, Haemophilus influenza, Streptococcus pneumonia, Streptococcus pyogenes, Acinetobacter calcoaceticus-baumannii, Enterobacter aerogenes-cloacae, Escherichia coli, Klebsiella pneumoniae, proteus* spp., *Pseudomonas aeruginosa, Serratia marcescens, Staphylococcus aureus*).Tests for other viral infections (i.e., cytomegalo virus (CMV); Epstein–Barr virus (EBV); herpes simplex virus (HSV); varicella-zoster virus (VZV).Detection of the viral vector of the COVID-19 vaccine.

## 3. Results

### 3.1. Structure of the AEFI Procedures for COVID-19 Vaccination

The proposed procedures are based on the same structure of the WHO AEFI guidelines, including eligibility (case ascertainment), checklist, algorithm and final classification (Figure 1).

### 3.2. Eligibility

Eligibility represents an important prerequisite that triggers the next step in the assessment of causality. Usually at this stage, the reviewers defined the so-called “causality question”. Considering the severe adverse effects reported in the literature, in the case of the COVID-19 vaccination, the question is if the COVID-19 vaccine/vaccination caused thrombosis and thrombocytopenia. 

### 3.3. Checklist

To explore the possible causal association between the COVID-19 vaccination and severe adverse effects, several questions were included in the checklist (Table 1). The table has been organized into four sections: (1). Order of the evidence; (2). temporal proximity; (3). evidence for other causes; (4). published evidence. The first step is the definition of a temporary relationship between COVID-19 vaccine administration and the onset of adverse effects. The second step is to analyze the temporal proximity to confirm that the symptoms occurred within a plausible time window after vaccine administration. According to the literature, three weeks after administration represent an ideal period that should be monitored to evaluate the AEFI after COVID-19 vaccination [8,9].

The third step is the evidence examination to exclude the presence of other causes. This step may be subdivided into six subsections: (a) history; (b) patient’s condition before symptoms; (c) patient examination; (d) immunization anxiety; (e) vaccine quality; and (f) immunization errors. Moreover, in order to better clarify the presence of any other evidence, it is suggested to insert any suspicious cause underlying the adverse effects. The checklist represents important guidance: following each item, it is possible to collect fundamental information about the conditions of the subject involved in the AEFI event.

In the last part of the checklist, it is important to report up-to-date evidence, based on peer-reviewed studies, on the causal association between COVID-19 vaccines and severe adverse effects. The evidence could either support or contradict the causal association.

### 3.4. Laboratory Tests and Results

The information about the laboratory tests was obtained analyzing the medical records of each case. When the tests were not reported, we performed the analysis in different samples (hospitalization samples, fresh/fixed tissue after autopsy, cadaveric biological fluids). We summarize the main data carried out for each case in Table 2.

### 3.5. Autopsy

The novelty of the proposed structure of the causality assessment for AEFI following COVID-19 vaccine administration is represented by the autopsy tool. In our opinion, it is important to stress the importance of the post-mortem investigation in the application of the algorithm. Autopsy is a fundamental diagnostic technique in cases of new or unfamiliar human pathologies [38]: in the same manner, it represents the gold standard method to clarify unknown diseases related to vaccine administration. In Italy, the Sicily Region with a specific act established a protocol to be applied in suspected cases of death (“Regional taskforce for the implementation of diagnostic findings and autopsy examinations in COVID-19 positive corpses or in the event of post-vaccination deaths”) [27].

Autopsy should be considered mandatory in all deaths temporarily related to vaccine administration. Performing a complete post-mortem investigation and sharing the data are very helpful steps to guarantee safe vaccination procedures.

### 3.6. Algorithm and Classification

Following the checklist step-by-step, it is possible to collect specific information in order to define causality between vaccination and severe adverse effects. As previously described, it is necessary to answer five questions, considering the first question a pre-requisite: the identification of an AEFI following administration of a COVID-19 vaccine is necessary to proceed with the causality assessment algorithm. As reported in Figure 2, the green answer reinforces the causality association between the two events, whereas the red answer implies poor causal association. 

At the end of the checklist, it is important to classify the event as follows: The AEFI is related to vaccine administration (in the presence of clear evidence of vaccine administration, confirmed by a temporal relationship, after the exclusion of other causes. Moreover, there is published evidence that confirmed the relationship).The AEFI is probably related with vaccine administration (in the presence of clear evidence of vaccine administration, confirmed by a temporal relationship, after the exclusion of other causes, but there is no published evidence that confirmed the relationship).The AEFI could be related to vaccine administration (in the presence of clear evidence of vaccine administration, confirmed by a temporal relationship. The presence of other causes that could be related to symptoms).The AEFI could not be related to vaccine administration (in the presence of clear evidence of vaccine administration, there is no clear temporal relationship; moreover, the causal association is not clear).The AEFI is not related to vaccine administration (there is no evidence of vaccine administration).

Finally, the case could be classified as “indeterminate”, when there is missing information in the checklist, with the impossibility to classify the case. When this case occurs, it is mandatory to collect additional data.

### 3.7. Application

To verify the relationship between the AEFI and COVID-19 vaccine administration, we performed the application of causality assessment guidelines adapted to COVID-19 vaccination for the first time. In particular, for each reported case, we applied the proposed checklist, and subsequently, the relative algorithm in order to determine a draft AEFI classification, according to one of six categories, defining if the case may be classified as “indeterminate”, “definitely non related”, “unlikely related”, “possibly related”, “probably related”, and “definitely related”. The application of the post-mortem examination helps us to achieve the final classification optimally.

As previously described, the innovation of the proposed model is the application of the post-mortem examination to define the exact cause of death; in this manner, the relative algorithm to achieve the final classification can be applied optimally.

The discussed cases concerned two subjects, one male and one female, in apparent previous healthy status. Moreover, they presented two similar events (thrombocytopenia, thrombosis, and brain hemorrhage) with lethal consequences. For all these reasons, both cases presented the prerequisite to suspect an AEFI after COVID-19 vaccination.

Following the checklist reported in Table 1, the first step is the evaluation of the “order of incidence”, answering affirmatively the following question: “Was the COVID-19 vaccine administered before the observed symptoms occurred?”.

The second step concerns the evaluation of temporal proximity: in case 1, the symptoms and death (“exitus”) of the patient occurred within 21 days following vaccination (symptoms after 10 days, death after 16 days); in case 2, the symptoms occurred 10 days after vaccination, and death after 24 days, even if she remained in a comatose state for 13 days. For these reasons, it is possible to conclude that the suspected AEFI is temporarily related to the COVID-19 vaccine administration.

To exclude the presence of other causes for the observed symptom, we analyzed each category (history; patient’s condition before the appearance of symptoms; examination of the patient; immunization anxiety; vaccine quality; immunization errors) for both cases. In the history section, both patients had never tested positive for COVID-19 infection. This data was confirmed through chemiluminescence immunoassay on samples collected during hospitalization (no IgM nor IgG antibodies were found). Moreover, no anti-COVID-19 vaccination had been administered before the event, and both subjects had never experienced diseases after any type of vaccination during his/her life. Analyzing the patients’ conditions before the appearance of symptoms, the clinical history was negative for all items (see Table 1, Section 3.4, Section 3.5, Section 3.6 and Section 3.7).

As previously described, the post-mortem investigation allows to collect evidence, playing a pivotal role for the causality assessment. The autopsy was performed in both cases following the recommended guidelines [24,26] and, carrying out complete histological and IHC investigations (Table 3). 

Moreover, in order to exclude preexisting coagulation disorders a complete test panel was performed (see boxes 3.11 and 3.12): no pathological findings were reported. The two cases reacted positively for antibodies to the platelet factor 4 PF4/heparin complex.

Considering that the swab sample could be a bias in the determination of COVID-19 positivity [25], in order to confirm negativity to SARS-CoV-2 infection, the molecular test was performed on lung swabs for case 1, and on bronchial alveolar lavage for case 2. Moreover, the RT-PCR molecular test for SARS-CoV-2 was performed on intestine FFPE samples in order to exclude virus persistence in the intestinal tissue following infection [39,40]. All samples tested negative in both cases. Furthermore, the panels both for respiratory viruses and respiratory bacterial pathogens tested negative in both cases, as well as the molecular test for the identification of adenovirus used as a viral vector for the administered COVID-19 vaccine (ChAdOx1 nCoV-19).

Analyzing other parameters (presence of stress response to vaccination, vaccine quality, presence of immunization errors), both cases were negative for all questions.

Finally, several international reports have been published highlighting the possibility of thrombohemorrhagic complications within the time interval of 5 to 16 days after the first dose of the adenoviral vector vaccine ChAdOx1-19 against COVID-19 [7,8,9,18]: these data confirmed that the analyzed cases may be definitively classified as AEFI caused by vaccination.

## 4. Discussion

Rare cases of unusual thrombotic events in combination with thrombocytopenia have been reported since the end of February 2021 after ChAdOx1 nCoV-19 vaccination. 

The lesson “to learn from the dead” [41] should be considered a rule and not only an opportunity to diagnose post-mortem unexplained deaths, especially when it is temporally related to vaccine administration. In these cases, the autopsy tool represents the gold standard method to gain all information about death [38]. The same approach could be useful in order to diagnose other morbid conditions in other disciplinary contexts (i.e., blood transfusion) [42]. Two kinds of diseases were reported: a rare form of cerebral venous sinus thrombosis (CVST) and cases of pulmonary embolism and splanchnic vein thrombosis associated with thrombocytopenia.

Based on the EMA report on 7 April 2021, there were 169 cases of thrombosis in the cerebral veins and 53 cases of thrombosis in the abdominal veins, with 18 fatal cases [6].

The EMA, PRAC (Pharmacovigilance Risk Assessment Committee), is responsible for evaluating all aspects of risk management of drug products for human use. The monitoring phase includes several steps such as recognition, assessment, risk reduction, and communication of adverse reactions [6].

Specifically, in the case of ChAdOx1 nCoV-19, PRAC analyzed 62 cases of CVST and 24 cases of splanchnic vein thrombosis from the authorization until 22 March 2021. Based on this report, the majority of the cases occurred in women under 60 years of age, usually within 2 weeks from the first dose. Moreover, PRAC concluded that to date it is very difficult to establish the exact cause of this phenomenon, but it is thought that the vaccine may generate an immune response leading to an atypical heparin-induced-thrombocytopenia-like disorder [6].

In order to ascertain the possibility that the vaccine may be considered a trigger for AEFI, each hypothesis for other causes should be excluded. To achieve this goal, we have suggested a complete checklist (Table 1) as guidance for clinical and forensic purposes. Moreover, we have reported an algorithm as guidance to define the causality assessment.

According to the discussed algorithm, in the first case, the patient suffered only from sleep apnea. Twelve days later the first administration of ChAdOx1 nCov-19 vaccination, he was admitted to the emergency department for portal vein thrombosis. The patient underwent a throat swab for SARS-CoV 2 for 3 times during his hospitalization period (total length of hospitalization was 4 days). No allergic reactions were reported in previous vaccinations. A complete autopsy was performed and showed extensive portal thrombosis. Laboratory tests were also performed (thrombosis panel test, genetic thrombophilia tests, molecular test for post-mortem detection of SARS-CoV2 in typical lung and atypical intestine tissues, and a complete panel for other infections by microorganisms). All data were negative. However, the search for anti PF4/polyanion-antibody gave positive results. Histological analysis in H&E confirmed portal vein thrombosis and immunohistochemical analysis was strongly positive for C1r, C4d and ICAM. Vaccine preparation and administration errors were excluded because it was the only case of adverse reaction with the same vaccine batch.

In the second case, the patient went to the emergency department 10 days after vaccination in an unconscious state. Her previous medical history was negative. She tested negative to a throat swab for SARS-CoV 2 twice, during her hospitalization period (total length of hospitalization 26 days). No allergic reactions were reported in previous vaccinations. A complete autopsy was performed, confirming the presence of a very large intracranial hemorrhage and of an occlusive thrombus in the superior sagittal sinus. Based on medical records and on the laboratory tests, all results were negative, with the exception of anti PF4/polyanion-antibody that showed positive results. H&E analysis confirmed right cerebral sinus thrombosis and immunohistochemical analysis was strongly positive for C1r, C4d and ICAM. Vaccine preparation mistakes and administration errors were excluded because it was the only case of adverse reaction with the same vaccine batch.

Based on the proposed workflow (Figure 2), the last step is the presence of literature in support of causal effects between vaccine administration and severe AEFI. The first hypothesis is based on previous studies conducted on subjects who had suffered from similar thrombotic events. Indeed, the clinical features of the subjects involved in vaccine adverse effects resemble a well-described disease, the so-called heparin-induced thrombocytopenia (HIT). The pathogenesis of this disease is related to the occurrence of antibodies versus platelet factor 4 (PF4, a factor involved in blood clot formation). The anti-PF4 antibodies that belong to the IgG class may also be produced by patients not receiving heparin therapy, probably caused by exposure to substances of polyanionic nature (such as molecules involved in the vaccine composition). It is important to note that our cases confirmed this theory: indeed, the presence of anti-PF4 antibodies was always detected. 

To distinguish HIT from this new syndrome, the term VITT, meaning vaccine-induced thrombotic thrombocytopenia has been formulated. The other mechanism to explain thrombus formation related to a reduction in the number of platelets (thrombocytopenia) had been previously described both during infections (COVID-19, influenza) and also following the administration of other types of vaccines (polio, pneumococcus, influenza, MPR-mumps, mumps, rubella and hepatitis). In these cases, the cause could be a cross-reaction with platelets of the new antibody. Indeed, following the aforementioned natural infection or vaccine administration, the antigens expressed on the surface of platelets or their precursors could be similar to the antigens of the pathogens (this phenomenon is known as molecular mimicry). As a result of this ’cross-reaction, platelets undergo lysis mediated by T lymphocytes or are phagocytosed by the ’scavenger’ cells of the spleen [7,8,9]. Moreover, it is also not excluded that platelets are directly involved in the synthesis of the spike protein after vaccine administration, inducing a mechanism of the autoimmune response against the same cells [43,44,45,46,47,48]. In the exemplificative cases herein presented, all tests performed in biological samples were negative.

The other hypothesis to explain the described cases is the so-called heparin-induced thrombocytopenia is that this event generates disseminated intravascular coagulation (DIC). DIC is characterized by the systemic activation of blood coagulation, which leads to the formation of intravascular thrombin and fibrin, with subsequent thrombosis of small and medium-sized vessels and finally severe hemorrhage due to coagulation factor and platelet consumption and activation of fibrinolysis [49]. This clinical picture is similar to the exemplificative cases, considering that we have excluded preexisting coagulation alterations. Intriguingly, the dramatic inflammatory hypercoagulability mechanism is similar to that of patients with a poor prognosis for COVID-19 [50]. This is not completely surprising because the observed picture belongs to the general mechanism of antibody directed enhancement, where paradoxically, the severe form of the disease shares aspects with vaccine adverse events [51,52]. On this basis, now the open question is if FP4-dependent platelet activation could also be involved in the disease pathophysiology.

Therefore, other hypothesis should be explored to explain the observed adverse effects. First, the existence of undiagnosed SARS-CoV-2 infection should be excluded, considering that the coagulation problems are very similar to the COVID-19 patients [25,38,53,54,55,56,57]. This hypothesis is excluded in the two reported cases, considering that we performed several molecular investigations in different tissue samples, including atypical tissues (i.e., intestine).

Another alternative hypothesis could be the possibility of impurity in some components of the vaccine. This hypothesis was excluded in the two cases considering that two different batches were used.

Finally, the role of the vector should be better investigated, specifically a chimpanzee adenovirus, testing the post-mortem samples (lung, spleen, and cadaveric blood) for the presence of adenovirus vectors of administered vaccine (ChAdOx1 nCoV-19). This investigation tested negative in both cases. 

Based on the discussed evidence, we may define a causality relationship between COVID-19 vaccine administration and AEFI, and the pathogenesis of this disease is related to the development of antibodies versus platelet factor 4. Particularly, as previously described, vaccination with ChAdOx1 nCov-19 may cause the rare development of immune thrombocytopenia mediated by platelet-activating antibodies against PF4, which clinically mimics heparin-induced autoimmune thrombocytopenia.

Although we discuss here with two exemplificative cases related to ChAdOx1 nCoV-19 vaccination, we strongly recommend using this diagram in all cases of AEFI related to COVID-19 vaccination, considering that severe adverse effects could occur with any kind of vaccine.

## 5. Conclusions

Causality evaluation of AEFIs is not only crucial to counteract current vaccine hesitancy and suspicion, but also for implementing an international evidence-based vaccination policy. Because each AEFI has distinctive properties, the scientific community is called on to develop a specific checklist for each vaccine. In this paper, we provide a basic framework applicable to the causality assessment of AEFI that occurr after COVID-19 vaccination.

Although we developed this methodology in Italy, we expect that it will be generalizable to other countries because it has been developed on the basis of the WHO framework, revision of international literature, and additional evidence. A systematic and regular revision by different expertise is desirable, and it should be supported by governments. Moreover, considering the novelty of COVID-19 vaccines, the adoption of standard procedures could be useful to gain important information about the great challenge of the current century.

## Figures and Tables

**Figure 1 diagnostics-11-00955-f001:**
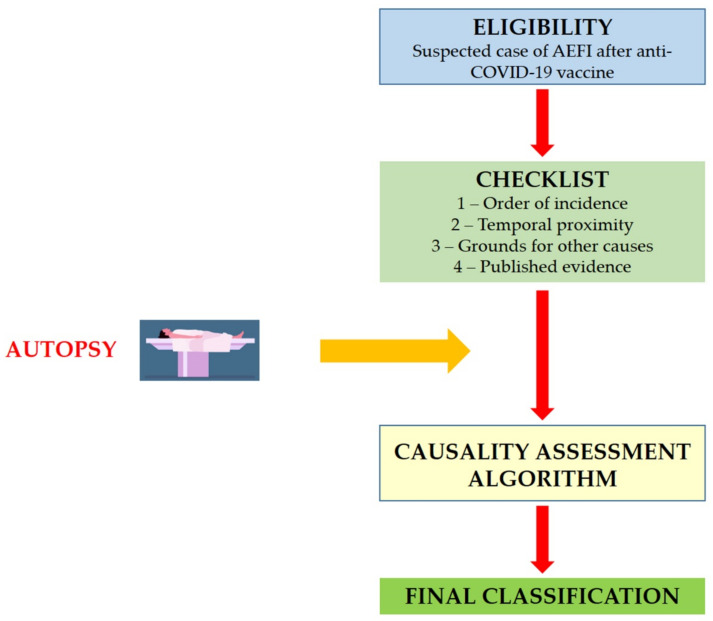
Structure of the causality assessment guidelines for AEFI following COVID-19 vaccine administration.

**Figure 2 diagnostics-11-00955-f002:**
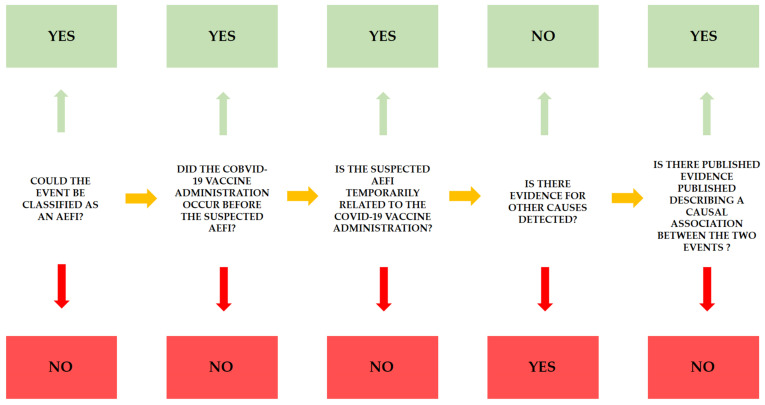
For each AEFI, this flowchart should be applied in order to define the causality assessment between the two events, vaccine administration and AEFI. The green answers reinforce the relationship.

**Figure 3 diagnostics-11-00955-f003:**
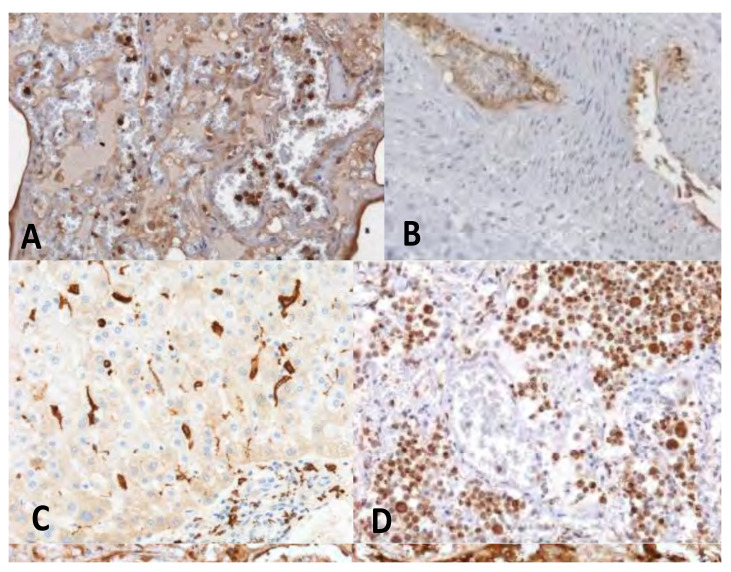
IHC in the exemplificative cases of VITT demonstrated the activation of the innate immunity and of the complement pathway at the level of vascular and perivascular tissues of the major organs. (**A**) lung stained for the complement fraction C1r (20×); (**B**) Cephalic vein with deep vein thrombosis stained for the complement fraction C4d (10×); (**C**) liver positive for CD 163 cells infiltrates (20×); (**D**) clusters of activated CD66 cells in the lung (20×).

**Table 1 diagnostics-11-00955-t001:** Causality assessment guidelines for AEFI following COVID-19 vaccine administration: checklist.

**1. Order of events**							
**Was the COVID-19 vaccine administered before the observed symptoms occurred?**						**Yes**	**No**
**2. Temporal proximity**							
**Did the observed symptoms occur within 21 days following vaccination?**						**Yes**	**No**
**Write the time interval between vaccination and occurrence of observed symptoms**						Insert time in days	
**3. Evidence for other causes**							
**• For each question, please answer Y (Yes), N (No), UK (Unknown), NA (Not applicable)** **• Please write the reason for your choice in the note column.**							
**Category**	**Items**	**Y**	**N**	**UK**	**NA**	**Note**	
**History**	3.1 Has the patient ever tested positive for COVID-19 infection? Please, write the time and the duration of COVID-19 infection						
	3.2 Has the patient ever been vaccinated for COVID-19 infection before? If so, please write the name of vaccine and the vaccination date.						
	3.3 Has the patient ever experienced any diseases after any type of vaccination?						
**Patient’s condition before appearance of symptoms**	• Has the patient ever experienced any items below before the manifestation of COVID-19 vaccine severe adverse effects? If so, please indicate the event data.						
	3.4 Upper respiratory infections						
	3.5 Personal history of other venous thrombosis and embolism (ICD-10- CM Z86 Diagnosis Code)						
	3.6 Severe cardiomyopathy (ICD-10-CM category I42)						
	3.7 Other infections (i.e., HBV, HCV, HIV)						
	3.8 Surgery						
	3.9 Other pathologies						
**Patient examination**	• Was a test among those listed below performed after the manifestation of COVID-19 vaccine severe adverse effects? If so, write the date of the test, test result, and specific other useful information.						
	3.10 Post-mortem investigation (with histological and IHC investigations						
	3.11 Thrombosis panel test						
	3.12 Genetic testing for hypercoagulability						
	3.13 Molecular tests for detection of SARS-CoV-2 using other lung samples						
	3.14 Molecular tests for detection of SARS-CoV-2 in atypical samples, using formalin-fixed paraffin-embedded (FFPE) tissue specimens						
	3.15 Panel for respiratory viruses						
	3.16 Panel for respiratory bacterial pathogens						
	3.17 Tests for other viral infections						
	3.18 Detection of viral vector COVID-19 vaccine						
**Immunization anxiety**	3.19 Were the observed symptoms a stress response to vaccination? (e.g., acute stress response, vasovagal syncope, hyperventilation, anxiety, etc.)						
**Vaccine quality**	3.20 Could the vaccine given to this patient have a quality defect or be substandard or counterfeit? (i.e., checking production lot, storage condition, etc.)						
**Immunization errors (please, write type of error, if any)**	3.21 Did anything unusual occur during vaccination preparation? (e.g., incorrect mixing, use of expired vaccine, abnormal physical condition, etc.)						
	3.22 Did anything unusual occur during the vaccination procedure? (e.g., Inoculation timing/dose/site/route, needle size error, etc.)						
**• If there are any suspicious causes other than those listed above, write the details below.**							
**4. Published evidence (literature, WHO GACVS, IOM etc.) regarding a causal association between the vaccine and observed symptoms.**							

**Table 2 diagnostics-11-00955-t002:** Summary of main tests performed relative to each proposed case. The data obtained from medical records are not reported.

Test	Method	Sample		Result	
		Case 1	Case 2	Case 1	Case 2
Genetic testing for hypercoagulability (Factor V Leiden Mutation, Factor II 20210 Mutation)	Real-time PCR (CVD6 MULTIPLEX REAL TIME, Nuclear Laser Medicine s.r.l., Settala (MI), Italy)	Hospitalization sample (blood), Cadaveric Blood, Cadaveric Tissue (spleen)	Hospitalization sample (blood),	Wild type	Wild Type
Molecular tests for detection of SARS-CoV-2 using other lung samples	Real-time PCR (Allplex™ SARS-CoV-2 Assay, Arrow Diagnostics, Genova, Italy)	Cadaveric lung swab	Bronchial wash sample	Negative	Negative
Molecular tests for detection of SARS-CoV-2 in atypical samples, using Formalin-Fixed Paraffin-Embedded (FFPE) tissue specimens	Real-time PCR (following the described protocol by Facchetti et al. [36])	FFPE Tissues (lung and intestine tissues)	FFPE tissues (lung and intestine tissues)	Negative	Negative
Detection of viral vector COVID-19 vaccine virus	Real-time PCR (following the described protocol by Rohr et al. [37])	Cadaveric Tissues (lung, spleen, cadaveric blood)	Cadaveric Tissues (lung, spleen, cadaveric blood)	Negative	Negative
IgG Anti-heparin/PF4	Automated chemiluminescent anti-heparin/PF4 immunoassay	Hospitalization sample (blood)	Hospitalization sample (blood)	Negative 0.0.2 U/mL (>1 U/mL positive)	Negative 0.21 U/mL (>1 U/mL positive)
IgG Anti-polyanion/PF4	Anti PF4/polyanion complex (PF4 Enhanced Test, Immucor, Waukesha, WI, USA).	Hospitalization sample (blood)	Hospitalization sample (blood)	Positive	Positive

**Table 3 diagnostics-11-00955-t003:** Main findings of post-mortem examination.

	Case 1 (Male, 50 y.o.)	Case 2 (Female, 37 y.o.)
**Macroscopic findings**	Portal vein thrombosis with smaller thrombi in the splenic and upper mesenteric veins. Intracranial hemorrhage in the subarachnoid region.	Occlusive thrombus in the superior sagittal sinus and a very large hemorrhage in the frontal cerebral lobe. Moreover, in the axillary region of the left arm, a thrombus was detected
**Microscopic findings** **(H&E staining)**	The microscopic evaluation revealed numerous vascular thrombi and intense hemorrhagic phenomena localized in the meningeal space and extravasated in the brain tissue. The thrombotic phenomena involved small and medium vessels most likely due to damage of their walls, which induced endothelial activation and an inflammatory reaction with a procoagulant process and thrombotic reaction.	
**IHC findings** **(Figure 3)**	Immunohistochemistry showed at the level of the vascular and perivascular tissues of heart, lung, liver, kidney, ileum and deep veins, the expression of adhesion molecules (VICAM1) and activated inflammatory cells (CD66b+, CD 163, CD 61+) expressing the complement fraction C1r. At the endoluminal level the inflammatory cells appeared to be arranged in clusters with aggregated platelets. Finally, a massive deposition of immunoglobulins of the IgM and IgG classes was apparent in the same vascular and perivascular locations (Figure 3).	

## Data Availability

The data presented in this study are available on request from the corresponding author. The data are not publicly available due to privacy restrictions.

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
