# Peer review of "COVID-19 Vaccine and Death: Causality Algorithm According to the WHO Eligibility Diagnosis"

_diagnostics, 2021, doi:10.3390/diagnostics11060955_

Round 1

Reviewer 1 Report

I read with great interest the article entitled "Covid-19 vaccine and death: causality algorithm according to the WHO eligibility diagnosis".

In their work the authors address a current and debated topic: the side effects of vaccines with particular reference to thrombosis / thrombocytopenia.

However, the paper presents several critical issues starting from the title. It is unclear whether they want to discuss a single vaccine (ChAdOx1 nCoV-19 was administered in both cases) or vaccines in general.

The authors then set themselves different objectives:

  1. a) illustrate the applicability of the WHO protocol for ascertaining the case-effect link between vaccines and thrombosis
  2. b) describe the cases they observed
  3. c) elucidate the pathophysiologic mechanism of adverse events following immunization (AEFI).

However, none of the three were fully satisfied.

In the event that after the revision the result is not considered for publication, I recommend dividing the work into 2/3 papers.

 The different parts of the paper will be analyzed below.

1 Introduction: The WHO guidelines (WHO Causality Assessment Of An Adverse Event Following Immunization; 2^ Edition.; 2019; ISBN 9789241513654.) talk about the evaluation of the causal link not of pathogenesis. I do not consider the elements presented in any way sufficient to clarify this aspect from an experimental point of view. Authors should represent the possibilities in an exclusively hypothetical way and with updated literature.

2.3 Vaccine characteristics and exhaustive review of the literature

The literature should be updated.

2.3(?-The sub-chapters should be numbered consecutively.) Exemplificative cases needing of diagnosis of AEFI. The cases concerned exclusively concern patients vaccinated with (ChAdOx1 nCoV-19) this data should be reported.

2.3 (?The sub-chapters should be numbered consecutively.)  Autoptic Methodology. The authors should specify what was the time interval between death and the execution of the autopsy assessment and the sampling of biological fluids.

 If the autopsy methodology follows a precise protocol, reporting it in full is redundant. Authors should refer to the protocol applied, possibly highlighting the reasons for which there was a deviation. It should be noted that portal vein thrombosis was described in one of the cases. This pathology is rare and is generally due to various factors (Liver disease, malignancy, thrombophilic factors, inflammatory state and myeloproliferative disorders. Some of the predisposing pathologies (hepatopathies) were excluded with autopsy or - as for the presence of thrombophilic factors with other exams. How did you rule out the presence of a tumor or a myeloproliferative pathology? And that of an inflammation? The algorithm for the indications provided by the specific case should be implemented.

2.3 (…) Tests and Biomarkers the data reported are completely insufficient.

  1. a) Authors should first specify the substrate used and its nature (cadaver sampling? Hospitalization choir sampling). b) Secondly, the methods used to carry out the tests should be specified (including brand, operating principles, aliquot of sample used and CUTT-OFF of the single test).
  2. c) Last but not least, in the event that the origin of the sample is cadaverous, has the use of the applied tests been validated for cadaveric samples?

3.5 (…) Application

The authors do not report the results of the tests which they claim to have carried out other than a brief summary table which is not useful to trace the conditions of the cases examined.

The authors report: "Analyzing the patients' conditions before the appearance of symptoms, the clinical history was negative for all indications items ...". such an analysis? Is it based on objectivity or on the story of a third party? Are there instrumental tests?

  1. Discussion

Although the vaccine is the same principle for clarity the authors should give the same name to the same in the case description (line158 ) and in the discussion (line 441).

Always in the discussion instead of discussing the results obtained or the functionality of the flow-chart, the authors speak of the pathogenesis of vaccine thrombosis not based on original data. The text must be completely rewritten because it goes beyond what is presented in the rest of the paper (among other things, a chapter reserved for current knowledge has been introduced in a chapter of the text).

I suggest to the authors to make a selection of the reported data in order to give greater clarity to their work. Although currently the paper is not absolutely considerable for publication, considering the novelty of the topic and the debate on it, I believe it is right to give the authors the opportunity to improve it by making the appropriate changes.

Author Response

REVIEWER 1

I read with great interest the article entitled "Covid-19 vaccine and death: causality algorithm according to the WHO eligibility diagnosis".

In their work the authors address a current and debated topic: the side effects of vaccines with particular reference to thrombosis / thrombocytopenia.

However, the paper presents several critical issues starting from the title. It is unclear whether they want to discuss a single vaccine (ChAdOx1 nCoV-19 was administered in both cases) or vaccines in general.

RESPONSE: Although in this paper we report the experience of two cases of fatal adverse effects related to ChAdOx1 nCoV-19 vaccine administration, we suggest applying the proposed workflow to any severe adverse effects possibly related to the COVID-19 vaccination. Indeed, all proposed steps could be useful to establish the causal effects between any vaccine administration and fatal cases. We have better elucidated this important aspect in the study's aims.

The authors then set themselves different objectives:

  1. a) illustrate the applicability of the WHO protocol for ascertaining the case-effect link between vaccines and thrombosis
  2. b) describe the cases they observed
  3. c) elucidate the pathophysiologic mechanism of adverse events following immunization (AEFI).

However, none of the three were fully satisfied.

In the event that after the revision the result is not considered for publication, I recommend dividing the work into 2/3 papers.The different parts of the paper will be analyzed below.

RESPONSE: The aims of the present paper are now reduced and more clearly stated, starting from the abstract.  

1 Introduction: The WHO guidelines (WHO Causality Assessment Of An Adverse Event Following Immunization; 2^ Edition.; 2019; ISBN 9789241513654.) talk about the evaluation of the causal link not of pathogenesis. I do not consider the elements presented in any way sufficient to clarify this aspect from an experimental point of view. Authors should represent the possibilities in an exclusively hypothetical way and with updated literature.

RESPONSE: Thank you for the opportunity to clarify this important aspect. We have now rewritten the aims of the study at lines 101-108.

2.3 Vaccine characteristics and exhaustive review of the literature

The literature should be updated.

RESPONSE: We have now added, after the reference list, appendix 1, which contains an exhaustive and updated review of the WHO documents about anti-Sars-Cov2 vaccines.

2.3(?-The sub-chapters should be numbered consecutively.) Exemplificative cases needing of diagnosis of AEFI. The cases concerned exclusively concern patients vaccinated with (ChAdOx1 nCoV-19) this data should be reported.

RESPONSE: Following your suggestion, we have renumbered all sections consecutively. Moreover, we have changed the title of subsection 2.4 as reported below:

"Exemplificative cases needing AEFI diagnosis after ChAdOx1 nCoV-19 vaccine administration".

Finally, it is important to remark that fatal adverse effects after vaccine administration are uncommon: for this reason, we have discussed two managed cases. Moreover, it is important to state that for the first time we describe the results of autopsy cases of AEFI 

.

2.3 (?The sub-chapters should be numbered consecutively.)  Autoptic Methodology. The authors should specify what was the time interval between death and the execution of the autopsy assessment and the sampling of biological fluids.

RESPONSE: We have inserted this missed information at line 183. “Before the autopsy, both medical records were carefully consulted. Moreover, family medical histories were obtained from the respective general practitioners. In case 1, the patient suffered only from obstructive sleep apnea (OSA). The autopsy was performed 24 h after death. In case 2, personal medical history was nil. In this case, autopsy was performed 3 days after death; the corpse was stored at −4 â—¦C to avoid post-mortem modification. All biological fluids collected during the hospitalization period were properly stored and were used to perform all tests.”

 If the autopsy methodology follows a precise protocol, reporting it in full is redundant. Authors should refer to the protocol applied, possibly highlighting the reasons for which there was a deviation. It should be noted that portal vein thrombosis was described in one of the cases. This pathology is rare and is generally due to various factors (Liver disease, malignancy, thrombophilic factors, inflammatory state and myeloproliferative disorders. Some of the predisposing pathologies (hepatopathies) were excluded with autopsy or - as for the presence of thrombophilic factors with other exams. How did you rule out the presence of a tumor or a myeloproliferative pathology? And that of an inflammation? The algorithm for the indications provided by the specific case should be implemented.

RESPONSE: We have inserted this information.

“Furthermore, before each autopsy, a total body CT scan examination was performed in order to confirm the absence of tumors.” Line 186.

“For the first time, we adopted the protocol established by Sicily’s Regional task-force for the implementation of diagnostic findings and autopsy examinations in COVID-19 positive corpses or in the event of post-vaccination deaths.” Line 229.

“Finally, to exclude a myeloproliferative disease, a bone marrow biopsy at the left iliac crest was performed before the autopsy examination. Moreover, a myeloproliferative neoplasm is likely to be excluded by the negative history thereof and normal blood counts in repeated tests routinely done before the event in both subjects. Line 257.

2.3 (…) Tests and Biomarkers the data reported are completely insufficient.

  1. a) Authors should first specify the substrate used and its nature (cadaver sampling? Hospitalization choir sampling). b) Secondly, the methods used to carry out the tests should be specified (including brand, operating principles, aliquot of sample used and CUTT-OFF of the single test).
  2. c) Last but not least, in the event that the origin of the sample is cadaverous, has the use of the applied tests been validated for cadaveric samples?

 RESPONSE: We thank the reviewer for the opportunity to clarify this important aspect. In the hospitalization period, different tests were performed in both patients. These are a part of the blood examination performed during the hospitalization period.

Thrombosis panel test

Reference value

Additional Information

PT- Ratio

0.8-1.2

Instrument: ACL TOP 750 LAS - Instrumentation Laboratory (Werfen)

Method: Chromogenic and immunological assay

Aliquot of sample: 50 μL

INR

therapeutic range

aPTT

24-36 sec

aPTT Ratio

0.8-1.2

D-Dimer

0-700 ng/mL FEU

Antithrombin

80-120%

Protein C

70-140%

Protein S

55-124%

Lupus Anticoagulant Testing

dRVVT Screening

dRVVT Confirm

 SCT

< 1.2

< 1.2

< 1.2

Antiphospholipid Antibodies

aB2-GP1 IgG

aB2-GP1 IgM

ACA-IgG

ACA-IgM

< 20

< 20

< 20

< 20

Instrument: ACL Acustar  - Instrumentation Laboratory (Werfen)

Method: Chemiluminescent assayAliquot of sample: 50 μL

ID-Heparin/PF4 antibody test

≥ 0.400 Optical Density (OD)

Methods: ELISA assays (Immucor)

Aliquot of sample: 50 μL

HIT-Ab(PF4-H)

> 1 U/mL: Present

< 1 U/mL: Absent

Instrument: ACL TOP 750 LAS - Instrumentation Laboratory (Werfen)

Method: Chromogenic and immunological assay

Aliquot of sample: 50 μL

HIT-IgG(PF4-H)

> 1 U/mL: Present

< 1 U/mL: Absent

Instrument: ACL Acustar  - Instrumentation Laboratory (Werfen)

Method: Chemiluminescent assayAliquot of sample: 50 μL

Abbreviations: anti-B2-GlycoProtein 1 (aB2-GP1); AntiCardiolipin antibodies (ACA); Dilute Russell Viper Venom Time (dRVVT); HIT: Heparin-Induced Thrombocytopenia (HIT); INR: International Normalized Ratio; PF4-H: Platelet Factor 4-Heparin; PT: Prothrombin Time; aPTT: Partial Thromboplastin Time; SCT: Silica Clotting Time.

Complete Blood Count (CBC)

Parameter

Reference value (Male)

Reference value(Female)

Units

White Blood Cell

4-11

4-11

103/µL

Neutrophils %

40-74

40-74

%        

Lymphocytes %

20-48

20-48

%

Monocytes %

3-11

3-11

%

Eosinophils %

0-8

0-8

%

Basophils %

0-1.5

0-1.5

%

Absolute Neutrophils count

2-8

2-8

103/µL

Absolute Lymphocytes count

1-5

1-5

103/µL

Absolute Monocytes count

0.16-1

0.16-1

103/µL

Absolute Eosinophils count

0.02-0.8

0.02-0.8

103/µL

Absolute Basophil count

0-0.2

0-0.2

103/µL

Red Blood Cell

4.2-5.5

3.8-5

106/µL

Hemoglobin

12-18

12-16

g/dL

Hematocrit

37-52

35-48

%

MCV

80-99

80-99

fL

MCH

26-32

26-32

pg

MCHC

32-36

32-36

g/dL

RDW%

11-15

11-15

%

RDW

38-48

38-48

fL

Platelets

150-450

150-450

103/µL

Abbreviations: MCV: Mean Corpuscular Volume; MCH: Mean Corpuscular Hemoglobin; MCHC: Mean Corpuscular Hemoglobin Concentration; RDW: Red cell Distribution Width.

Many other examinations were performed. Obviously, we inserted in the main text only the tests that may be functional to the proposed flowchart; in addition, we have inserted a new table (TABLE 3).

3.5 (…) Application

The authors do not report the results of the tests which they claim to have carried out other than a brief summary table which is not useful to trace the conditions of the cases examined.

RESPONSE: We have inserted this missed information, inserting a new table (TABLE 3).

The authors report: "Analyzing the patients' conditions before the appearance of symptoms, the clinical history was negative for all indications items ...". such an analysis? Is it based on objectivity or on the story of a third party? Are there instrumental tests?

RESPONSE: We have inserted this missed information.

  1. Discussion

Although the vaccine is the same principle for clarity the authors should give the same name to the same in the case description (line158 ) and in the discussion (line 441).

Response: Thank you for this suggestion. We have modified it.

Always in the discussion instead of discussing the results obtained or the functionality of the flow-chart, the authors speak of the pathogenesis of vaccine thrombosis not based on original data. The text must be completely rewritten because it goes beyond what is presented in the rest of the paper (among other things, a chapter reserved for current knowledge has been introduced in a chapter of the text).

Response: Thank you for this suggestion. We have improved and reorganized the discussion.

I suggest to the authors to make a selection of the reported data in order to give greater clarity to their work. Although currently the paper is not absolutely considerable for publication, considering the novelty of the topic and the debate on it, I believe it is right to give the authors the opportunity to improve it by making the appropriate changes.

Reviewer 2 Report

Authors presented two Italian cases of deaths as adverse events following immunization, after vaccination against SARS-CoV-2. They used guidelines of WHO for presentation of cases. I suggest some corrections:

  1. In Introduction should be described also that many other vaccines are during clinical trials. Please cite the following article https://pubmed.ncbi.nlm.nih.gov/33408775/, in which are data about other vaccines. Authors should also write about other adverse effects observed after vaccinations.
  2. In Material and Methods, should be add names of reagents, tests, kits and their producers.

Author Response

Reviewer 2

Authors presented two Italian cases of deaths as adverse events following immunization, after vaccination against SARS-CoV-2. They used guidelines of WHO for presentation of cases. I suggest some corrections:

  1. In Introduction should be described also that many other vaccines are during clinical trials. Please cite the following article https://pubmed.ncbi.nlm.nih.gov/33408775/, in which are data about other vaccines. Authors should also write about other adverse effects observed after vaccinations.

RESPONSE: Thanks for this suggestion. We have inserted this missed citation.

  1. In Material and Methods, should be add names of reagents, tests, kits and their producers.

RESPONSE: We have inserted this missed information, inserting a new table (TABLE 3).

Reviewer 3 Report

I want to thank the Editor for giving me the opportunity to review the Diagnostics article. Pomara et al. In my opinion they carry out an extensive and important research work on a very current topic in the scientific field about “thrombosis and thrombocytopenia after COVID-19 vaccination”. The study design shows a correct and rigorous methodological imprint despite the difficulty of the topic and the complexity of the data available, the basic indications are given following the basic framework of the World 45 Health Organization (WHO). For these reasons I believe that it is an article that deserves publication and will be of interest to the readers of Diagnostics and will allow its use in clinical practice and a possible academic debate by allowing the article to be cited.
The consideration is particularly valuable: 500-503 "The lesson“ to learn from the dead ”[41] should be considered a rule and not only an opportunity to diagnose post -mortem unexplained deaths, especially when it is temporally related to the vaccine administration. In these cases, the autopsy tool represents the gold standard method to gain all information about death [38]. " In fact, this approach is necessary for scientific correctness in terms of causation and in forensic medicine. The same approach could be useful in order to diagnose other morbid conditions present in other disciplinary contexts such as blood transfusion and its lesser known risks as it is also indicated in this work: doi: 10.1016 / j.transci.2020.102779. This aspect could be mentioned.
Also a small clarification might be useful: lines 433-434 the authors stated "This data was confirmed during the post-mortem investigation, considering that no IgM nor IgG antibodies were found in the post-mortem samples." Please, insert the test used to evaluate the presence of IgM and IgG antibodies. Moreover, clarify if this test was performed on cadaveric blood on hospitalization samples.
Having solved these small changes, I believe that the work has already been affected by substantial and useful changes to improve the text, therefore, in my opinion, it can be published.

Author Response

20th May, 2021

Dear Editor

We thank the referees for their time and for their helpful comments improving the scientific quality of our manuscript. Please find below a copy of the point-by-point response to the reviewers’ comments.

Thanking you again for all your help and consideration.

Reviewer #3:

I want to thank the Editor for giving me the opportunity to review the Diagnostics article. Pomara et al. In my opinion they carry out an extensive and important research work on a very current topic in the scientific field about “thrombosis and thrombocytopenia after COVID-19 vaccination”. The study design shows a correct and rigorous methodological imprint despite the difficulty of the topic and the complexity of the data available, the basic indications are given following the basic framework of the World 45 Health Organization (WHO). For these reasons I believe that it is an article that deserves publication and will be of interest to the readers of Diagnostics and will allow its use in clinical practice and a possible academic debate by allowing the article to be cited.

  1. Thank you for the positive comments. Following the reviewer’s suggestion, we have improved the manuscript.

The consideration is particularly valuable: 500-503 "The lesson“ to learn from the dead ”[41] should be considered a rule and not only an opportunity to diagnose post -mortem unexplained deaths, especially when it is temporally related to the vaccine administration. In these cases, the autopsy tool represents the gold standard method to gain all information about death [38]. " In fact, this approach is necessary for scientific correctness in terms of causation and in forensic medicine. The same approach could be useful in order to diagnose other morbid conditions present in other disciplinary contexts such as blood transfusion and its lesser known risks as it is also indicated in this work: doi: 10.1016 / j.transci.2020.102779. This aspect could be mentioned.

Thanks for this suggestion. We have inserted this important consideration with the relative reference. Lines 460-62

Also a small clarification might be useful: lines 433-434 the authors stated "This data was confirmed during the post-mortem investigation, considering that no IgM nor IgG antibodies were found in the post-mortem samples." Please, insert the test used to evaluate the presence of IgM and IgG antibodies. Moreover, clarify if this test was performed on cadaveric blood on hospitalization samples.

  1. Thanks for this suggestion. We have clarified this important information. Lines 392-394

Having solved these small changes, I believe that the work has already been affected by substantial and useful changes to improve the text, therefore, in my opinion, it can be published.

  1. Thank you for your positive comments.

Please do not hesitate to contact me for any further questions.

Round 2

Reviewer 1 Report

Despite the changes made, the work still has very important limits that can hardly be corrected.. It should be noted in particular:

Line 160. Compared to the previous version of the paper, it emerges that a CT would have been performed to assess the presence of unknown tumors. Is this method validated in postmortem? Although there are studies limited to a small number of patients, this procedure is not codified. It is also noted that the interpretation of CT data in the cadaver requires the evaluation of a professional who is familiar with this type of examination in postmortem. What type of CT was used? What is his sensitivity? The authors are able to add some CT images taken into consideration that would enrich the work.

Line 167 Would the authors have taken blood samples from critically ill patients and “stored” it for posthumous tests including genetic ones? The data should be clarified because otherwise the study would show ethical limits that cannot be overcome. Specimen storage methods should be clarified in particular if substances have been added or particular vials used for this purpose.

The authors once again did not clarify the nature of the blood used, brand and sensitivity of the instruments and methods used for the described determinations. This constitutes another major drawback of the study. In case of non-validation of the methods for cadaveric samples, the result of “negativity”, in particular in cadavers with a three-day PMI, is not to be considered reliable, especially considering the physiological hematic alterations in postmortem.

This limit is particularly evident for the following determinations: "Thrombosis panel test (D-dimer, prothrombin time and international normalized ratio (PT / INR), partial thromboplastin time (PTT, aPTT), complete blood count (CBC), antiphospholipid antibodies, lupus anticoagulant testing, antithrombin, protein C, protein S, and ID-Heparin / PF4 antibody test. ”This type of investigation should be carried out for points 5,6,7,8.

A table like the one prepared for a small part of the tests (Table 2) should be prepared for such exams.

Line 239 Materials and methods as regards iliac crest sampling (also not described in the previous version) should be explored by citing adequate literature. Once again the authors should clarify whether a diagnosis on a withdrawal of this nature has been validated for postmortem.

In conclusion, the method proposed by the authors presupposes an important battery of examinations in the postmortem (acceptable) but it must necessarily be corroborated by a necessary slice of information coming from samples taken in the living (not acceptable if there is no indication to carry out the test or a specific approved experimental project).

On the basis of these data, all previous proposals should be adapted. At present, I do not believe that the authors can make changes that would make the paper suitable for publication.

Author Response

15th May, 2021

Dear Editor

We thank the referees for their time and helpful comments that are helping to improve the scientific quality of our manuscript. After the first revision step, we had answered point-by-point to the reviewers’ comments. We are surprised that one of them is not satisfied with our modifications and that he stated that “Despite the changes made, the work still has very important limits that can hardly be corrected.”. Nevertheless, we are grateful to the reviewer for the opportunity to better clarify several important aspects of this manuscript, that on our opinion is of great interest to the scientific community, in particular for the worldwide efforts in the vaccination program.

RESPONSES TO REVIEWERS

Reviewer #1:

Line 160. Compared to the previous version of the paper, it emerges that a CT would have been performed to assess the presence of unknown tumors. Is this method validated in postmortem? Although there are studies limited to a small number of patients, this procedure is not codified. It is also noted that the interpretation of CT data in the cadaver requires the evaluation of a professional who is familiar with this type of examination in postmortem. What type of CT was used? What is his sensitivity? The authors are able to add some CT images taken into consideration, would enrich the work.

As we have reported in the main text (lines 188-189 for case 1; lines 196-197 for case 2), during the hospitalization period both patients underwent CT examination: no sign of cancer was found. Moreover, analyzing the medical records for each patient, no suspect of tumors was raised by their different doctors during hospitalization. Furthermore, the use of the CT scan post-mortem confirmed the negative data collected during the hospitalization period. Finally, we did not understand why the reviewer did not accept that “This finding was confirmed by the detailed post-mortem examination of all organs.” For all these reasons, we apologize but fail to understand the reviewer’s criticisms: all these data were reported in the main text. Indeed, we have written that their family doctors reported that each patient did not suffer from any tumors. Moreover, during the hospitalization period, nobody suspected nor confirmed the presence of tumors, notwithstanding numerous blood tests and instrumental data. Finally, our post-mortem investigation was negative (both instrumental and histological). So, we have difficulty to understand the reviewer’s reasons and believe that they are unjustified. Anyway, we have inserted as requested the type of CT scan used.     

Line 167 Would the authors have taken blood samples from critically ill patients and “stored” it for posthumous tests including genetic ones? The data should be clarified because otherwise the study would show ethical limits that cannot be overcome. Specimen storage methods should be clarified in particular if substances have been added or particular vials used for this purpose.

This criticism is unjustified. First of all, both autopsies were forensic autopsies. We stated that “The prosecutor authorized the use of anonymous data according to the Italian law”. For this reason, the use of stored samples is always allowed.  Moreover, we used the stored samples only for tests not performed. We have inserted (following the suggestion of the reviewer) in table 2 all tests performed and the type of samples. Finally, we have declared that “All biological fluids collected during the hospitalization period were properly stored”: it means that, for example sera or plasma were stored at -20°C.

The authors once again did not clarify the nature of the blood used, brand and sensitivity of the instruments and methods used for the described determinations. This constitutes another major drawback of the study. In case of non-validation of the methods for cadaveric samples, the result of “negativity”, in particular in cadavers with a three-day PMI, is not to be considered reliable, especially considering the physiological hematic alterations in postmortem.

This criticism is unjustified. We have inserted (following the suggestion of the reviewer) in table 2, all the information about methods. Another important aspect is related to the validation of the methods used and applied to cadaveric samples. First of all, we refer to several important papers that we have inserted as references in the main text (i.e. https://doi.org/10.3390/jcm9072026; https://doi.org/10.3390/diagnostics10080575). These papers reported that in post-mortem investigations lung swabs are more reliable than the troth swabs for the diagnosis of SARS-CoV-2 infection. Moreover, we have stated in the main text that post-mortem samples were used to confirm negativity of this infection previously reported in the medical records. Finally, we suggest to consider the novelty of these adverse effects: so we have difficulty to understand the critique, because several methods are validated both for our in vivo and in post-mortem investigations (for example the research of adenovirus vaccine vector).

This limit is particularly evident for the following determinations: "Thrombosis panel test (D-dimer, prothrombin time and international normalized ratio (PT / INR), partial thromboplastin time (PTT, aPTT), complete blood count (CBC), antiphospholipid antibodies, lupus anticoagulant testing, antithrombin, protein C, protein S, and ID-Heparin / PF4 antibody test. ”This type of investigation should be carried out for points 5,6,7,8. A table like the one prepared for a small part of the tests (Table 2) should be prepared for such exams.

This criticism is unjustified. We have clearly written that many blood tests were performed during the hospitalization period. We are in the position to describe all methods used but it is clear that to do this we shall need to produce a laboratory book. In all publications on COVID-19, nobody has inserted in Material and Methods sections the detailed description of each laboratory test acquired from medical records. Nevertheless, we would be able to do it but the text would became unreadable, missing the goal of this paper: THE APPLICATION OF CAUSALITY WORKFLOW. THESE ARE ONLY TWO EXEMPLIFICATIVE CASES.

Line 239 Materials and methods as regards iliac crest sampling (also not described in the previous version) should be explored by citing adequate literature. Once again the authors should clarify whether a diagnosis on a withdrawal of this nature has been validated for postmortem.

We have modified the text inserting this sentence: “To confirm the absence of a myeloproliferative neoplasm, a bone marrow biopsy at the left iliac crest was performed before the autopsy examination”. As described above, all data collected from their family doctors were negative for these diseases. Moreover, during the hospitalization period no data supported this hypothesis.

In conclusion, the method proposed by the authors presupposes an important battery of examinations in the postmortem (acceptable) but it must necessarily be corroborated by a necessary slice of information coming from samples taken in the living (not acceptable if there is no indication to carry out the test or a specific approved experimental project).

The reviewer missed the importance of our take-home message: this workflow should represent guidance in all similar cases. First of all, it is important to examine carefully medical records Moreover, when a forensic autopsy is required, all kinds of investigations can be performed. Perhaps the reviewer ignore the possibility to perform all kinds of laboratory tests in order to ascertain the truth in the context of the forensic autopsy.

On the basis of these data, all previous proposals should be adapted. At present, I do not believe that the authors can make changes that would make the paper suitable for publication.